



# Short-term effects of fertilization on dissolved organic matter (DOM) in soil leachate

Alexandra Tiefenbacher[1], Gabriele Weigelhofer[2,3], Andreas Klik[4], Matthias Pucher[2,3], Jakob Santner[5], Walter Wenzel[6], Alexander Eder[1] and Peter Strauss[1]

[1]Federal Agency for Water Management, Petzenkirchen, Austria
[2]Institute of Hydrobiology and Aquatic Ecosystem Management, University of Natural Resources and Life Sciences, Vienna, Austria
[3]WasserCluster Lunz GmbH, Austria
[4]Institute for Soil Physics and Rural Water Management, University of Natural Resources and Life Sciences, Vienna, Austria
[5]Institute of Agronomy, University of Natural Resources and Life Sciences, Tulln, Austria
[6]Institute for Soil Research, University of Natural Resources and Life Sciences, Tulln, Austria

*Correspondence to*: Alexandra Tiefenbacher (alexandra.tiefenbacher@baw.at)

**Abstract.** Besides the importance of dissolved organic matter (DOM) in soil biogeochemical processes, there is still a debate on how agricultural intensification affects the composition and concentration of dissolved organic matter leached from soils

into adjacent aquatic ecosystems. In order to investigate the immediate response of DOM leaching to fertilization, we conducted a short-term (45 day) lysimeter experiment with undisturbed silt loam and loamy sand soil cores. Mineral (calcium ammonium nitrate) or organic (pig slurry) fertilizer was applied on the soil surface with a concentration equivalent to 130 kg N ha$^{-1}$. After fertilization, soil leachate was collected in 6-days intervals. Dissolved organic carbon concentrations (DOC) were measured with gas chromatography, while shifts in DOM composition were analysed using absorbance and

excitation- emission fluorescence indices from peak-picking as well as from PARAFAC analysis.

During the first 12 days, fertilization of a silt loam reduced DOC concentrations in the leachate and shifted its composition towards more microbial- like compounds. Additionally, the discrepancy in DOM composition between fertilizer and control treatments of a silt loam increased with time. However, in loamy sand only mineral fertilization affected organic matter leaching and decreased DOC concentrations in the leachate during the first 12 days. Furthermore, mineral fertilization of the

loamy sand led to DOM compounds with low molecular size in the first 12 days. Our results show that fertilization tends to increase microbial transformed DOM, while it reduces leached DOC concentrations. Furthermore, the magnitude of fertilization on DOC concentrations and DOM composition was highly depending on the soil texture they originate from. However, in our set-up, the experimental soil units were restricted to a soil depth of 16 cm (Ap horizon). At ecosystem level, a sufficiently long soil passage might mitigate the impact of fertilization on soil DOM.



## 1 Introduction

Soil stores twice as much carbon as the atmosphere (Davidson and Janssens, 2006), with an estimated soil organic carbon storage of 1462 to 1548Pg in the first 100 cm of the soil profile (Batjes, 2014). Soil carbon storage is increasingly threatened by the conversion of land into areas for intensive agricultural production. By now, already 37.7 % of the global terrestrial surface area is used for agricultural production (FAO, 2014). However, in an effort to improve global food security for an exponentially growing human population, the conversion of land to agricultural areas is expected to increase further, and the production per square meter is predicted to intensify (Curtis et al., 2002). The latter, intensive production, is often achieved through the application of nitrogen-based fertilizers, with a predicted annual application of 118 million tons of N globally by 2020 (FAO, 2015). Decades of generously dimensioned fertilizer application rates have increased the overall nitrogen content in soils, thus fuelling mineralization rates which, in turn, enhanced the depletion of naturally occurring soil organic carbon (Jones et al., 2012). This has implications for the receiving stream and river ecosystems. Land conversion has increased carbon transport from soils to aquatic systems from approximately 1.1 Pg y$^{-1}$ to 1.9 Pg y$^{-1}$ globally (Regnier et al., 2013). These terrestrial carbon fluxes are mostly of organic origin (Battin et al., 2009), whereby the largest share is in dissolved form (dissolved organic matter, DOM; Alvarez-Cobelas et al., 2012).

Overall, the intensification of agricultural production has altered both the concentration and the composition of DOM in fluvial systems through the reduction of the soil organic matter pool, alterations of hydrological pathways due to extensive drainage networks, and changes of in-stream microbial processes resulting from increased nutrient and light availability (Graeber et al., 2012; Stanley et al., 2012; Wilson and Xenopoulos, 2009). Terrestrial DOM influences various biogeochemical processes in the aquatic system, controlling microbial respiration and nutrient uptake, amongst others, all of which are related not only to the DOM quantity, but also to its biodegradability and chemical reactivity (Bernhardt and Likens, 2002; Fischer et al., 2002; Yamashita and Jaffé, 2008; Tank et al., 2010; Derenne and Nguyen Tu, 2014). As a consequence, agriculturally derived DOM can alter the ecological state and the biodiversity of aquatic ecosystems severely (Fischer et al., 2002; Lipson, 2015). Therefore, it is imperative to know how and to which extent agricultural management practices, such as fertilization, may change the composition of terrestrial DOM discharged into streams.

During the last decade, several studies have evaluated the impact of agriculture on DOM composition in streams and/or catchments. Intensification of agriculture tends to increase protein-like, labile, and microbially-transformed organic matter with low redox states and low molecular weight and aromaticity (Wu et al., 2019b; Hosen et al., 2014; Wilson and Xenopoulos, 2009). However, in some studies, organic matter in agricultural streams exhibited highly complex structure with a high levels of humified and microbial-derived DOM (Graeber et al., 2012; Heinz et al., 2015). These partly contradicting results arise from the fact that aquatic DOM is shaped by the terrestrial DOM source, which depends on land use, soil texture, and agricultural management practices, the flow paths by which the DOM is transported to the stream (e.g., overland flow, groundwater, drainage water), and the various in-stream microbial transformation processes (Stanley et al.,



2012). In order to evaluate the potential impacts of fertilization on aquatic DOM composition, these different factors have, thus, to be analyzed separately.

Stabilization of DOM against leaching depends on the chemical nature and physical properties of sites available to interact

with the ions and molecules present in soil solution (Baldock and Skjemstad, 2000; Silveira, 2005). For instance, soil clay minerals and metal hydrous oxides (Vázquez-Ortega et al., 2014) are able to adsorb organic matter. The association between organic matter and inorganic soil colloids determines if enrichment or depletion of carbon is the dominating process (Qualls and Richardson, 2003). Consequently, distinct soil properties control which of the concurrent processes, stabilization or biodegradation, is dominating soil organic matter cycling. Stabilization of carbon may remove otherwise biodegradable

organic matter from solution. Soil organic matter can be protected from microbial decomposition in clay minerals by reducing its accessibility (Kravchenko et al., 2015). Additionally, the balance between carbon stabilization and biodegradation is controlling the organic matter content of the surface soils (Cleveland et al., 2004; Neff and Asner, 2001). Furthermore, leaching of nutrients and organic carbon is directly proportional to water flow. Soil texture is determining the water flow and water holding capacity, thereby controlling the leaching of dissolved nutrients in soils (Tiemeyer et al.,

75    2017).

Besides soil properties, fertilization is altering the pedogenic DOM composition by influencing the abundance as well as the activity of soil microorganisms. Especially nitrate is promoting microbial decomposition of DOM, which is indicated by positive correlations between nitrate content and aromaticity of the water extractable organic matter (WEOM) found in soils (Seifert et al., 2016). Furthermore, Fang et al. (2014) observed an increase in aromaticity and molecular complexity of

WEOM caused by mineral fertilization of an alpine meadow, while organic fertilization experiments conducted on forest soils did not affect WEOM compositions, according to a study by Ohno and Bro (2006). Moreover, in a study testing non-fertilized controls and mineral (NPK) fertilized samples of an Ultisol, an increase of molecular weight and higher ratio of microbially derived WEOM were observed, while organic fertilization had no impact on WEOM (Xu et al., 2018). One problem, however, is the current lack of a standardised method to obtain WEOM in soils. This is crucial since WEOM

concentration and its composition is affected by several factors, such as the extraction solution (water or diluted salt solution), soil pre-treatments (fresh or air-dried), soil solution ratios, and the contact time (De Feudis et al., 2017). In order to evaluate DOM leaching in soils under conditions as realistic as possible, systematic studies attempting to preserve soil aggregate structure while conducting their experiments are needed (Kravchenko et al., 2015; Tiemeyer et al., 2017). Consequently, we used undisturbed soil cores with different soil textures in lysimeter experiments to study the impacts of

fertilization and soil texture on the composition and concentration of leached DOM in soil pore water. We hypothesized that:

Both, mineral and organic fertilization will increase the concentrations of leached dissolved organic carbon (DOC) in the soil pore water and will shift DOM composition towards more labile compounds.

Pore water of coarser textured loamy sand will exhibit higher DOC concentrations than finer textured silt loam soil.

In order to evaluate the immediate response of soil DOM to fertilization, we established a short-term (45 days) lysimeter

experiment using undisturbed soil cores of different soil textures. Mineral (calcium ammonium nitrate) or organic (pig



manure) fertilizer was added on top of the soil surface with a concentration equivalent to 130 kg N ha$^{-1}$. In the following, soil cores were subjected to continuous rainfall simulated in an in vitro environment. Changes in the concentration of leached DOC and the composition of DOM were assessed in 6-day intervals using spectrophotometric and spectrofluorometric methods.

## 2 Material and Methods

### 2.1 Study sites and soil sampling

The study sites are representative of productive soil managed as agricultural crop land. For covering a high variety in soil texture, two contrasting areas were chosen. Firstly, the undisturbed Ap horizon (0- 16 cm) of a silt loam was sampled in Petzenkirchen, Lower Austria (49° 9'N, 15° 9'O; 323 m a.s.l.). The climate around Petzenkirchen can be classified as humid 105 with a mean annual temperature of 9.5 °C and a mean annual precipitation of 823 mm y$^{-1}$. The soil is characterised as a Cambisol (Working Group WRB, IUSS, 2015) cultivated with maize as the dominating crop. Secondly, the undisturbed Ap horizon (0-16 cm) of a loamy sand soil was collected in Rutzendorf, Lower Austria (48° 12'N, 16° 3'O, 156 m a.s.l.). This region can be described as semi-humid with a mean annual temperature of 9.5 °C and a mean annual precipitation of 529 mm y$^{-1}$. The soil is considered as a fine sandy Haplic Chernozem (Working Group WRB, IUSS, 2015) and agriculturally 110 managed with vegetables and cereals in the crop rotation.

For the lysimeter experiment, twelve undisturbed soil cores (Ap horizon; 0 - 16 cm, Ø 15 cm) were sampled at each study site. Due to the limited number of available lysimeters, sampling of the soil cores was performed sequentially. For ensuring similar initial conditions regarding the soil porosity, soil cores were taken three days after seedbed preparation. Silt loam soil was sampled in Petzenkirchen on 12/04/2017, while loamy sand soil was sampled in Rutzendorf on 24/10/2017. Due to the 115 high contribution of the intensively managed soil surface to microbial turnover (Kravchenko et al., 2015), we sampled the soil layers to the management depth of 0-16 cm.

Additional samples were taken to measure soil properties in two soil depths, ranging from 0- 8 and 8- 16 cm, respectively. Soil sampling for chemical analysis was bulked for each soil layer. Soil physical properties were measured in undisturbed samples using stainless steel cylinders (height 5 cm; volume: 250 cm³). In total, six samples for chemical analysis and six 120 samples for assessing soil physical soil properties were taken.

### 2.2 Experimental Design

The collected soil cores were amended with different types of fertilizer and then artificially irrigated. The factor fertilization consisted of three treatments: pig slurry ("organic"), mineral fertilization ("mineral") using calcium ammonium nitrate (NAC 27, 13.5% NH$_4^+$ and 13.5% NO$_3^-$, Borealis L.A.T GmbH, Linz, Austria) and no fertilization as a control ("control"). Applied 125 nitrogen concentrations equalled 130 kg N ha$^{-1}$, a typical Austrian application rate for maize and wheat (Richtlinien für die sachgerechte Düngung, 2017). The fertilizers were mixed with 50 mL deionized water and applied evenly on the soil

surface. In total, 0.235 g nitrogen were applied on the soil surface of each soil core. Each fertilization treatment was replicated 4 times for each soil texture, amounting to 12 cores per experimental soil in total, and the cores were randomly arranged to ensure the same microclimatic conditions. The experiment was conducted in two batches, from 28/06/2017 –

19/08/2017 for the silt loam and from 25/10/2017 – 14/12/2017 for the loamy sand. Each lysimeter batch was incubated for 45 days.

## 2.3 Lysimeter experiment

The lysimeter experiments were conducted under controlled environmental conditions of 17°C air temperature and 78 % relative humidity. Except during sampling, the cores were kept in the dark to minimize algae growth and photodegradation

of the DOM in the soil leachate samples. Immediately after fertilization, artificial raining of soil cores started with an intensity of 6 mm day$^{-1}$ using deionized water with a DOC content of < 0.5 mg L$^{-1}$ and electric conductivity of 1- 2 μS cm$^{-1}$. The chosen rainfall intensity was a result of taking into consideration a practical experimental time, sufficient amounts of leachate for the chosen soils and sufficient time to exchange soil water volumes. It corresponds to an event which typically reoccurs two to three times per year (eHyd, 2019). The duration of the experiment was set in a way that soil pore volume

was exchanged three times (soil pore volume: 1724 cm$^3$). This required 45 days and during the entire experiment each soil core received 5 litres of deionized water in total. An irrigation head with 18 mounted needles ensured even raining of the soil surface (Figure S 1). The pressure potential at the lower end of the soil cores was kept at – 60 hPa throughout the whole experiment, maintaining the soil moisture at field capacity, i.e. at 43 Vol% for the silt loam and 39 Vol% for the loamy sand, respectively. Soil water content and matrix potential within the soil cores were measured at 4 cm and 8 cm depth below the

soil surface employing ECH2O EC-5 small moisture sensors (Decagon Devices, WA, USA) and tensiometers (T5-10, METER Group AG, WA, USA).

Due to the low rain intensity, soil leachate had to be collected over longer time periods to obtain enough leachate for chemical analysis. Leachate samples for concentrations of TOC (total organic carbon), nitrate and chloride were collected over 6 days, resulting in a total of 450 ml leachate. Samples for assessing the DOM composition and DOC concentration

were collected for 16 h, resulting in a total of 80 ml leachate. During this collection, the bottles were kept cold with ice cubes at 4°C. The DOM samples were immediately filtered with pre-combusted GFF-filters (Whatman) and stored in pre-combusted glass vials in the dark at 4°C until analysis within 24 h.

## 2.4 Chemical Analysis/ Analytical measurements

### 2.4.1 Soil

Soil samples were air dried, sieved < 2 mm and analysed for pH (OeNORM EN ISO 10390, 2005), water content at sampling time (gravimetrically after drying at 105°C), total organic carbon (TOC, dry combustion, EN ISO 10694, 1995), total inorganic carbon (Scheibler procedure, DIN EN ISO 10693, 2014), bulk density (OeNORM EN ISO 11272, 2017) and





particle size distribution (sieving/pipette method, ISO 11277, 1998). Soil texture was classified according to WRB (Working Group WRB, IUSS, 2015).

Hydropedological characteristics, such as water retention curve and hydraulic conductivity function, of soil were analysed using the HYPROP® system (UMS, Germany). For this purpose, pressure heads within a soil sample (stainless- steel cylinders; height 5 cm; volume: 250 cm³) were measured continuously at two depths, supplemental water evaporation from soil surface was quantified by weighing (Schindler et al., 2010).

After the lysimeter experiment, soil cores were cut horizontally at a soil depth of 8 cm into two layers (0- 8 and 8 to 16 cm,

respectively). Total organic and inorganic carbon was analysed to obtain the remaining organic carbon of the specific soil layer.

### 2.4.2 Soil leachate

TOC and DOC concentrations were determined using gas chromatography (TOC-L, Shimadzu, Japan) following the procedure OeNROM EN 1484 (1997). Accuracy of DOC measurements is ± 5% up to 50 mg L$^{-1}$, based on repeated

measurements of potassium hydrogen phthalate standard solutions. Due to incomplete removal of dissolved inorganic carbon during the last 5 observations of the second experiment, DOC concentrations had to be predicted from the absorbance spectra for these samples by using a two- component model (Carter et al., 2012) obtained from absorbance at 254 nm and 350 nm (r$^2$=0.99). Chloride and nitrate concentrations were measured by ion chromatography (OeNORM EN ISO 10304-1, 2016; Dionex ICS- 900 Ion Chromatograph, Thermo Fischer; MA, US), while ammonia was determined photometrical

(Agilent 8453, CA, US) using the modified OeNORM L 1091 (1999) for water samples.

### 2.5 DOM characteristics

DOM composition was determined via optical characteristics using spectrophotometric and spectrofluorometric methods at WasserCluster Lunz GmbH (emission-excitation matrices EEMs; McKnight et al., 2001). Spectral absorbance was analysed at 0.5 nm increments between 200 and 700 nm in a 0.05 m quartz cuvette using a spectral photometer (UV1700 Pharma

Spec, Shimadzu Corporation, Kyoto, Japan Chin et al., 1994). DOC concentrations of these samples were analysed with a Sievers 5210C TOC-Analyzer.

Fluorescence spectra were measured using a fluorescence spectrophotometer (Hitachi F- 7000, Hitachi High- Technology Corporation, Tokio, Japan) coupled with a xenon lamp as a source for excitation. Fluorescence intensity was measured at excitation (ex) wavelengths ranging from 200 nm to 450 nm and emission (em) wavelengths ranging between 250 nm to 600

nm in 5 nm increments in a 0.01 m quartz cell (McKnight et al., 2001). All measurements were baseline-corrected using the Raman spectra of ultrapure water. Absorbance and fluorescence analyses were performed at constant room temperature of 21°C.

The evaluation of recorded fluorescence and absorbance spectra of DOM was done using the "staRdom" package with an inclusion of the "eemR"- package (Pucher et al., 2019) of the R- software 1.1.447 (R Core Team, 2019). The following



fluorescence indices were calculated: biological index (BIX; Huguet et al., 2009), 'Coble- Peaks' B, T, A, M and C, (Coble, 1996), fluorescence index Fi (McKnight et al., 2001) and humification index Hix (Ohno, 2002). Absorbance slope parameters SUVA$_{254}$ and E2:E3 were calculated after Helms et al. (2008) and Twardowski et al. (2004). The parallel factor analysis (PARAFAC) was used to link the fluorophores of the EEMs to chemical components in DOM (Murphy et al., 2013). Consistency of the PARAFAC analysis was maintained by checking randomness of residuals, visualisation of spectral

loadings as well as the half-split analysis. The random initialisation was set to 1000 iterations (Murphy et al., 2013).

## 2.6 Quality assurance and statistical analysis

The accuracy and precision of the measurements were monitored continuously by internal standards, laboratory replicates, matrix samples, reference material and standard solutions. The maximum allowed relative standard deviation between replicates was set to 5%.

All statistical analyses were performed using R 1.1.447 (R Core Team, 2019). The significance level for all statistical analyses was established at $p \leq 0.05$. For the analysis of changes in nutrient and DOC concentrations, 161 soil leachate samples in total were included in the statistics, with 4 replicates per combination of fertilizer treatment and soil texture and 7 observations during each experiment. Due to outlier correction, one silt loam soil core fertilized with mineral fertilizer had to be excluded from the analyses. All variables were tested for normality using a combination of Shapiro-Wilk Test (stats

package) and qq-plots (ggplot2 package). The homogeneity of the variances was verified using the Levene test (car package). Response variables were log-transformed to account for non-normal distribution when necessary. Influence of soil texture, fertilizer type and sampling date on hydrochemistry of the leachate water was tested using repeated measures analysis of variance (MANOVA; multcompView and lsmeas package). When Mauchly's test had been violated, the Greenhouse Geisser estimates of sphericity were used. If sphericity was met, multiple comparisons of means were calculated

with the Tukey's HSD test as post-hoc test (TukeyHSD; multcomp package).

For analysing the DOM composition, a total of 161 soil leachate samples were used allocated to 7 observations per experiment with 4 replicates per fertilization treatment and experimental soil. One soil core had to be excluded from the analysis due to outlier correction. Since prerequisites for parametric statistical evaluation could not be met, a non- parametric test was conducted. The main and interaction effects of soil texture and fertilizer type on the DOM spectra were evaluated

using a permutational MANOVA (PERMANOVA, vegan package, Euclidean distance, 999 iterations (Oksanen et al., 2013). Results of the PERMANOVA were visualised in Principal Response Curves (PRC; vegan package) for each soil texture individually in order to illustrate the impact of fertilizer type and their interaction with time on the DOM composition. PRC is a multivariate response technique based on repeated measures over time, the resulting graph allows to centralize the analysis on time-dependent treatment effects. As suggested by Van den Brink and Braak (1999), DOM indices with a species

weight between -0.5 and + 0.5 were not shown, because they were likely to show either a weak response or a response that is unrelated to fertilization and their interaction with time. The ANOVA-like permutation test (cca(), vegan package) assessed



the significance of the PRC model. Dunnet's test was applied to assess significant shifts in DOM composition over time, using the first sampling day as a reference (vegan package).

## 3. Results

### 3.1 Leachate chemistry

In general, both, soil texture and fertilizer type affected the concentrations of TOC, DOC, nitrate, and chloride leached from the soil cores (MANOVA, $p \leq 0.01$ and $\leq 0.001$, respectively; Table 2). Additionally, the effect of fertilizer type on TOC and DOC concentrations was interrelated with soil texture (MANOVA, $p \leq 0.05$ and $\leq 0.001$, respectively; Table 2). Overall, the effects were most obvious during the first 19 days, afterwards the leached nutrient and organic matter concentrations were more similar between the treatments and the control.

Average amounts of leached DOC over 45 days ranged between 20.4 mg (silt loam, mineral fertilizer) and 34.4 mg (loamy sand, organic fertilizer) (Figure 1). In silt loam, we observed significantly lower leached DOC amounts in the mineral fertilizer compared to the control (Tukey, $p \leq 0.01$, n= 21 per treatment). Significant differences were observed in loamy sand only between the organic fertilizer and the control (Tukey, $p \leq 0.05$, n=28 per treatment). Average amounts of leached TOC ranged between 22.3 mg (silt loam, mineral fertilizer) and 38.1 mg (loamy sand, organic fertilizer) and showed similar patterns as DOC. Total amounts of nitrate leached from the cores were highest in the mineral fertilizer treatment, followed by organic fertilizer and the control in both soil textures (Tukey, $p \leq 0.001$, for silt loam: n= 21 per treatment, for loamy sand: 28 per treatment; Figure 2). Compared to other treatments, only organic fertilization enhanced the total amounts of leached chloride at both soil textures (Tukey, $p \leq 0.001$, for silt loam: n= 21 per treatment, for loamy sand: 28 per treatment). On average, leached amounts of chloride amounted to 3, 1, and 35 mg in silt loam and 11, 18, and 48 mg in loamy sand for the control, the mineral, and the organic fertilizer, respectively.

Leachate percolating through unfertilized soil cores became gradually depleted in TOC, DOC, nitrate and chloride concentrations over time, until the concentrations stabilized in both experimental soils at approximately the same low to moderate levels after 19 days (Figure 1; Figure 2). Initial nitrate concentrations were higher in loamy sand than in silt loam (TukeyHSD, $p \leq 0.01$, for silt loam: n= 3 per observation, for loamy sand: 4 per observation, Figure 2), while loamy sand showed higher chloride concentration than silt loam during the first 12 days (TukeyHSD, $p \leq 0.05$, for silt loam: n= 3 per observation, for loamy sand: 4 per observation). In contrast, initial TOC and DOC concentration of unfertilized soil cores did not differ significantly between silt loam and loamy sand (TukeyHSD, p= 0.375, for silt loam: n= 3 per observation, for loamy sand: 4 per observation, Figure 1).

In silt loam, DOC concentrations in the leachate of fertilized soil cores were about half those of the control during the first 19 days and then increased to the same levels as the control (TukeyHSD, $p \leq 0.001$, n=3 per treatment and observation, Figure 1), independent of the fertilizer type. On average, leached DOC concentrations of the silt loam amounted to 10.6, 4.6 and 6.2 mg $l^{-1}$ on the first sampling date and to 8.0, 6.7 and 9.3 mg $l^{-1}$ after 24 days for the control, mineral and organic



fertilizer, respectively. In contrast to DOC, fertilization enhanced nitrate concentrations in the leachates of silt loam (Figure
2). In soil cores treated with mineral fertilizer, average nitrate concentrations in the leachate decreased continuously over the
whole experiment from 969 mg l$^{-1}$ at the beginning to 26 mg l$^{-1}$ after 45 days. In contrast, the initial nitrate concentrations of
organic fertilization were in the range of unfertilized soil cores with an average of 313 mg l$^{-1}$ and increased gradually
thereafter to a maximum of 463 mg l$^{-1}$ after 19 days. Afterwards, concentrations declined again to 30 mg l$^{-1}$. Chloride could
only be detected in silt loam treated with organic fertilizer during the first 12 days (TukeyHSD, p $\leq$ 0.001, n=3 per treatment
and observation).

In loamy sand, DOC concentrations were significantly lower in the treatment with mineral fertilizer than in the control
during the first 12 days (TukeyHSD, p $\leq$ 0.001, n=4 per treatment and observation, Figure 1). Afterwards concentrations
increased, reaching similar levels as the control (TukeyHSD, p= 0.384, n=4 per treatment and observation). Average DOC
concentrations from the organic fertilization were within the fluctuation range of the control throughout the experiment
(TukeyHSD, p= 0.669, n=4 per treatment and observation). Nitrate in fertilized loamy sand cores showed a reverse
development to DOC concentrations, with an initial increase after 12 days, followed by a fast decrease to values comparable
to the control (Figure 2). The highest nitrate concentrations occurred under mineral fertilization during the first 12 days
(TukeyHSD, p $\leq$ 0.001, n=4 per treatment and observation). In contrast, nitrate concentrations of organic fertilization were
similar to those of the control on the first sampling date (TukeyHSD, p $\leq$ 0.001, n=4 per treatment and observation), but
increased considerably until day 19 to values comparable to those of the mineral fertilizer treatment. After that, nitrate
concentrations declined in the same way as those in the treatments with mineral fertilization. Across all fertilizer types, only
organic fertilization resulted in a significant increase of chloride during the first 24 days (TukeyHSD, p $\leq$ 0.001, n=4 per
treatment and observation). Afterwards, concentrations in the leachate depleted rapidly and were in the range of the
unfertilized control (TukeyHSD, p= 0.497, n=4 per treatment and observation).

## 3.2 Composition of the leached organic matter

PARAFAC modelling of the Excitation- Emission matrix (EEM) identified six fluorophores (C1-C6) (Murphy et al., 2013).
Table 3 gives an overview about the observed components of PARAFAC modelling, their emission and excitation
wavelengths as well as the references they refer to. According to literature, fluorophores C1-C4 are similar to humic-like
fluorophores, whereby C1 and C2 are supposed to represent terrestrial DOM derived from forests and wetlands (Coble,
1996; Graeber et al., 2012; Romero et al., 2017), while C3 and C4 are probably microbially altered DOM, related to
agriculture and with a low molecular weight (Graeber et al., 2012; Tfaily et al., 2015). Fluorophore C5 is similar to protein-
tannin-like fluorophores (Romero et al., 2017) and C6 is assigned to microbially transformed soil fulvic acids (Williams et
al., 2010).

Soil texture, fertilizer type, and their interaction affected DOM composition of the soil leachate significantly
(PERMANOVA, p $\leq$ 0.001, Table 2). Correspondingly, soil texture explained 13% (R$^2$) of the variance. In addition,
composition of DOM shifted over time and sampling date descried 21% (R$^2$) of the variance. The interaction of soil texture



and time explained 28% ($R^2$) of the variance, forming the overall highest value, while the variance explained by fertilizer type and its interaction with time and soil texture was less than 10% (PERMANOVA, p < 0.001, Table 2).

The principal response curve (PRC) models calculated for DOM originating from silt loam explained 41% of the total variance composition data (PRC, p ≤ 0.001). For the silt loam, the first canonical axis of the PRC explained 93% of the total variance of the DOM composition (PRC, p < 0.001, Figure 3). This PRC axis revealed an interaction effect of fertilizer type with time (40%, PRC, p ≤ 0.001). Changes in DOM composition are displayed in the curve as well as in corresponding species weights (Figure 3). The disparity between DOM composition derived from the control and from fertilized silt loam cores increased with time, leading to an enrichment in humic-like (Peak A, M and C; C1, C3) and protein- like (Peak B and T; C5, C6) components of the control. In contrast to the aforementioned DOM indices, the fluorescence index FI (related to aromaticity), the freshness index BIX (indicator of freshly produced DOM from microbial activity), and the absorption index E2:E3 (related to molecule size) correlated positively with fertilization in silt loam (Figure 4). The E2:E3 ratio was significantly enhanced by fertilization during the first 19 days (TukeyHSD, p ≤ 0.05, n= 3 per observation), while BIX significantly increased between day 6 and day 18 (TukeyHSD, p ≤ 0.05, n= 3 per observation, Figure 4). Additionally, FI was significantly enhanced by fertilization in the very beginning (TukeyHSD, p ≤ 0.05, n= 3 per observation). However, none of the mentioned indices were affected by fertilizer type (TukeyHSD, p= 0.789, n= 3 per observation).

The principal response curve (PRC) models calculated for DOM originating from loamy sand explained 10 % of the total variance composition data (PRC, p ≤ 0.001). The first canonical axis of PRC explained 76 % of the total variance for DOM composition of a loamy sand (PRC, p ≤ 0.001, Figure 3). This axis revealed an interaction effect of fertilizer type with time (40%, PRC, p ≤ 0.001). Shifts in DOM composition as well as their species weights are displayed in Figure 3. C5, E2:E3, C6 and C2 were the most representative of the PRC 1 type response for the loamy sand soil. The effect of fertilization was limited to the first 12 days and occurred only during mineral fertilization. However, 86% of the total variance of DOM indices originating from loamy sand were described by time and only 10% of the total variance composition data could be described by fertilization (PRC, p ≤ 0.001, Figure 3). None of the Coble Peaks (peaks A, M, C, B, T) nor the humification index (HIX) or SUVA$_{254}$ responded to fertilization of the loamy sand. In general, FI, BIX, and E2:E3 responded positively to fertilization of loamy sand soil cores (Figure 4). Organic fertilization lead to enhanced BIX during the first 19 days, while mineral fertilization elevated E2:E3 ratio during the first 12 days (TukeyHSD, p ≤ 0.05 and < 0.01, respectively; n= 3 per observation). Additionally, fertilization increased FI during the first 12 days (TukeyHSD, p ≤ 0.01, n= 3 per observation), with no response to fertilizer type (TukeyHSD, p = 0.076, n= 3 per observation).

## 4. Discussion

### 4.1 Amounts of leached DOC

In contrast to our original expectations of enhanced DOC leaching from fertilized soils, fertilization reduced DOC concentrations of the leachate during the first 2-3 weeks. The only exception was organic fertilization of a loamy sand,



where similar DOC concentrations as in the control were observed throughout the whole experiment. Additionally, responses
of leached DOC concentrations to fertilization were time-depended and disappeared in the later phase of the experiment.
Over the entire experimental period, only mineral fertilization of the silt loam reduced the total amount of leached DOC,
while organic fertilization even enhanced the DOC concentrations in the leachate of the loamy sand. Irrespective of soil
texture, fertilization increased the leached nitrate concentrations during the first 24 days of the experiment. Furthermore, this
effect was fertilizer type specific and highest nitrate amounts were leached during mineral fertilization.

Fertilization affects DOC leaching in agricultural soils and the effect depends on the amount and type of applied fertilizer
(Bolan et al., 2011; Cleveland et al., 2004). In controlled laboratory experiments, continuous fertilization with $NH_4Cl$ and
$NaNO_3$ decreased DOC leaching of a mor humus (Oe and Oa) (Sjöberg et al., 2003). Despite the fact that physicochemical
properties of the O horizon used by Sjöberg et al. (2003) are different to our examined Ap horizon, results of reduced DOC
concentrations after mineral fertilization goes in line with our examination. Similar to our findings, $NH_4NO_3$ fertilization of a
tropical Oxisol reduced DOC concentrations in the soil leachate collected at 20 cm soil depth with zero tension lysimeters
(Lu et al., 2013). Both, Lu et al. (2013) and Sjöberg et al. (2003) concluded from their results that nitrogen- based
fertilization supported DOC mineralization, which in turn lead to a reduction of DOC concentrations in the leachate. In field
lysimeters, organic fertilization with pig manure enhanced leached DOC concentrations in a loamy clay soil (Long et al.,
2015). These authors assumed that the observed increase in DOC concentration was mainly caused by the high application
rate, which was 4.6 times higher than in our study (600 kg N ha$^{-1}$ y$^{-1}$). In another laboratory experiment with similar slurry
application rates than ours, organic fertilization did not affect DOC leaching of a silt loam, probably because the DOC was
rapidly mineralized by the microbial community as it percolated down the soil core (Lloyd et al., 2012). In deeper soil
horizons (up to 50 cm soil depth) DOC leaching is mainly affected by soil physical properties, such as adsorption and/or
flow velocity. Therefore, effects of N fertilization on the leaching of DOC decreased markedly with soil depth (Adams et al.,
2005; Don and Schulze, 2008). In addition, slower passage of the water down the soil profile enhances the contact time of
DOC with soil colloids, thus covering possible effects of fertilization on DOC leaching. In addition to fertilization, clay and
soil moisture content are main factors in controlling DOC concentrations in the soil pore water (De Troyer et al., 2014). Clay
is providing a large surface area for DOC sorption (Huang et al., 2019; Nguyen and Marschner, 2014; Singh et al., 2018),
and, sorption of DOC onto clay is mostly irreversible (Avneri-Katz et al., 2017). Due to the occupation of binding sites,
DOC sorption decrease with increasing organic carbon content (Bolan et al., 2011). Generally, soil texture is creating the
basis for soil water flow and water holding capacity, determining the leaching of dissolved organic carbon (Tiemeyer et al.,
2017). The relationship between water flow and water holding capacity is expressed by the hydraulic conductivity ($k_s$).
During our experiment, $k_s$ for loamy sand was higher than for silt loam. As a result, the loamy sand had a higher discharge
rate and DOC was leached more rapidly, reaching steady state conditions earlier than the silt loam. In contrast to the silt
loam, organic fertilization of the loamy sand caused an enhanced leaching of DOC, due to the combined effect of high
supply of organic matter, lower sorption site density, and higher flow velocity.



In our experiment the overall fertilizer effect on DOC leaching was limited to the first 2-3 weeks. Literature data also shows, that the fertilizer effect on DOC leaching is time depended. Immediately after fertilization, effects on DOC leaching could be observed (Lei et al., 2017; Long et al., 2015), while 4 month after fertilization no response was found (Wang et al., 2008).

In the field, the potentially leached organic matter content is generally estimated through soil water extracts. One problem, however, is the current lack of a standardised method to obtain water extractable organic matter (WEOM) in soil. This is crucial since WEOM is affected by several factors, such as the composition of the extractant (deionised water or diluted salt solution), soil pre-treatment (fresh or air-dried), soil solution ratios, and the contact time (De Feudis et al., 2017). Deviations between leached DOC concentrations sampled with zero tension lysimeters and the WEOM content have already been

discovered (Lu et al., 2013). Such deviations might be caused by overestimating the immobile fraction of organic matter, which is permanently or temporarily sorbed onto soil particles. Consequently, the impact of fertilization on DOC leaching may be difficult to detect by water extractable organic matter concentrations alone. In contrast to controlled laboratory experiments, the complexity of the system increases in *in-situ* field studies, offering a wide variety of competing processes that might produce and consume organic matter within soils. However, *in-situ* studies of DOC leaching with zero tension

lysimeters depend on the seasonal hydrological variability, which  may influence and even postpone the impact of fertilizer on the leached DOC concentrations (Don and Schulze, 2008; Kaiser and Kalbitz, 2012). Controlled laboratory experiments with soil columns as performed in our study have proved to be a promising method in analysing soil organic carbon leaching (Sjöberg et al., 2003; Lloyd et al., 2012). Further laboratory studies are needed to disentangle the complex interaction of the various abiotic and biotic factors controlling organic matter leaching from soils.

**4.2 Composition of leached organic matter**

In accordance with Zhang et al. (2019) and Xu et al. (2018), in our experiment humic-like components were the dominant fractions of leached DOM irrespective of soil and fertilizer type. Initial DOM originating from fertilized silt loam soil revealed a high aromaticity (Fi), a higher microbial activity (BIX) and a smaller molecular size (E2:E3) than the corresponding control. Additionally, the discrepancy in DOM composition between fertilized and control soils increased

with time, leading to enrichment in both humic-like and protein-like compounds in the control. However, in loamy sand only mineral fertilization affected DOM composition and, accordingly, DOM become enriched with microbial- like compounds though this effect was limited to the first 12 days. We assume that fertilization fueled the microbial decomposition of the soil organic matter, thus shifting DOM towards more labile compounds with low molecular weight and lower aromaticity. With time, leachate became gradually depleted in nitrate, the possibility to incorporate N into organic matter decreased, leading to

a loss of protein-like components in the leachate of fertilized soil cores. In the unfertilized control, nitrogen had to be mobilized from the soil organic fraction, which may explain the increase of protein-like fractions in the unfertilized soil over the duration of the experiment.

In literature, organic fertilization shifted the structure of DOM towards more labile compounds with a higher share of hydroxyl and amino groups, while mineral fertilization (NPK) did not qualitatively affect DOM composition (Ferrari et al.,

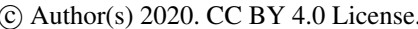



2011; Xu et al., 2018). Ferrari et al. (2011) suggested that long-term organic fertilization led to an enrichment of organic N, thus enhancing the lignin decomposition of crop residues. However, mineral fertilization resulted in an enrichment of hydrophilic compounds (McDowell et al., 2004) or enhanced labile organic compounds, such as carbohydrates, amino sugars, and proteins (Li et al., 2019). In a 56-days incubation experiment, mineral fertilization with $NH_4Cl$ accelerated microbial turnover of labile organic carbon, while the decomposition of recalcitrant soil organic matter declined (Zang et al.,

2017). Microbes might have lowered the decomposition of soil organic matter because their requirements were met by N fertilization (Zang et al., 2017). However, other studies showed that long-term fertilization can modify soil microbial decomposer communities, thus overlaying the effects of soil distinct properties and harmonizing DOM composition of soil humic substances (Li et al., 2019; Wu M. et al., 2019a).

Due to sorption onto clay minerals, soil passage causes a molecular fractionation of DOM. Aliphatic and low-polar

compounds are preferentially dissolved in the soil solution, whereas unsaturated, aromatic and highly polar compounds tend to be adsorbed onto soil particles (Huang et al., 2019). Furthermore, total soil nitrogen and organic carbon content increased by long-term fertilization is influencing the activity of the soil bacterial community, which in turn will affect soil DOM composition (Li et al., 2019). In our study even, the impact of fertilization was controlled by soil texture. Due to the higher discharge rate of the loamy sand, labile organic matter added as organic fertilizer was shortly retained in the soil core and

rapidly leached out, consequently no shift in DOM composition were observed.

In our experiment fertilizer-induced changes in DOM composition were time-depended and decreased with the continuation of the experiment. Labile organic compounds, such as simple carbohydrates, fats and amino acids, are degraded quickly during the first steps of decomposition. Other, more resistant organic substances such as cellulose, hemicellulose, and lignin are partially degraded and transformed at a lower rate. Therefore, soil carbon sequestration leads to stabilized mineral-

associated organic matter with slow/marginal microbial degradation and low mobility (Cleveland et al., 2004; Kaiser and Kalbitz, 2012).

In this experimental set up, responses of DOM leaching to fertilization were limited to a single fertilizer application. As shown before, long-term fertilization may have different effects. Additionally, the soil depth was restricted to 16 cm. At ecosystem level, a sufficient soil passage will shorten the effects of fertilization on aquatic DOM composition. However, this

experiment showed clearly that the effect of fertilization on DOC concentration and DOM composition was interrelated with soil texture. Future research should focus on the effects of shortening leaching pathways or soil drying over longer periods on the concentration and composition of leached DOM.

## 5. Conclusion

Due to the spatial extent of agricultural land use and the yearly growing application of nitrogen-based fertilizers, it is

reasonable to assume that the share of agricultural derived DOM will increase in terrestrial and aquatic ecosystems. Our study showed, that fertilization tends to increase microbial transformed DOM, while it reduces leached DOC concentrations.



The only exception was organic fertilization of a loamy sand, where leached DOC concentrations and DOM composition were equal to the unfertilized control. The impact of fertilization on DOC concentration and DOM composition was limited to the first two weeks after application. Besides, the experimental soil units were restricted to soil depth of 16 cm in our set-

up. At ecosystem level, a sufficient soil passage to aquatic structures might mitigate the impact of fertilization on fluvial DOM. Nonetheless, in overland flow, agriculturally derived DOM is entering aquatic systems without passing the soil passage. Therefore, it is crucial to know, whether and to which extent agricultural land management is altering terrestrial DOM.

On a global perspective, intensification of agricultural management may favour the leaching of biogeochemically reactive

DOM to aquatic ecosystems. Additionally, in future, many regions will become dry during summer with more precipitation per rainfall event, due to the global warming. High precipitation rates on dry soils usually generate a rapid surface runoff with a high sediment load. Therefore, we assume more agricultural derived DOM is entering aquatic systems without a sufficient soil passage in the future.

**Competing interests.** The authors have no conflicts of interest to disclose.

**Acknowledgements.** This study is an integral part of the project "Organic carbon cycling in streams: Effects of agricultural land use" (www.organic-carbon.at), which focuses on the impacts of agricultural land use on the concentration and composition of terrestrial DOM imported into a stream ecosystem via different flow paths as well as on the microbial

processing of terrestrial DOM within the aquatic system. The overall project was funded by the Provincial Government of Lower Austria (https://www.nfb.at/) within the Science Call 2015.



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





**Figure 1: Mean (± SD) dissolved organic carbon (DOC) in the soil leachate: A: Mean (± SD) total DOC amount: a summation of all time series within each treatment (left: loamy Sand; right: Silt Loam); statistical significance of difference between treatments within the soil textures are indicated with letters (Tukey; p < 0.05; Silt loam: n=21, loamy sand: n=28). B+C: temporal response curve of the mean (± SD) DOC concentration in the soil leachate percolating through Silt Loam (B, n= 3) and Loamy Sand (C, n=4); Shapes indicating different fertilizer application: control (circle); mineral (triangle) and organic (square).**

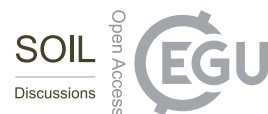

720

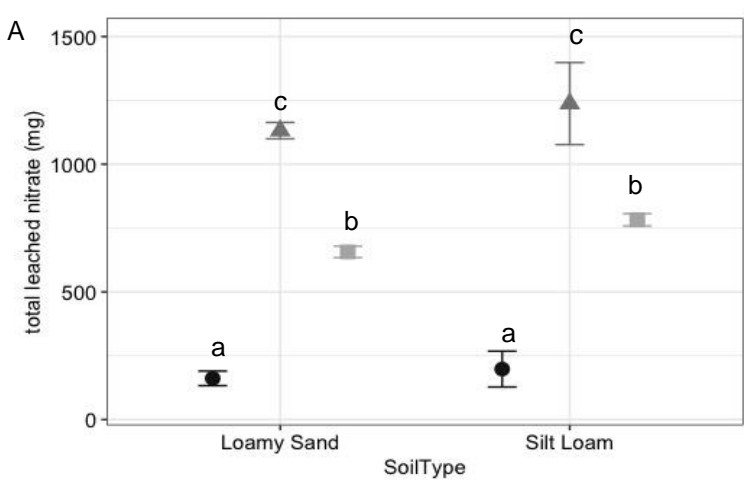

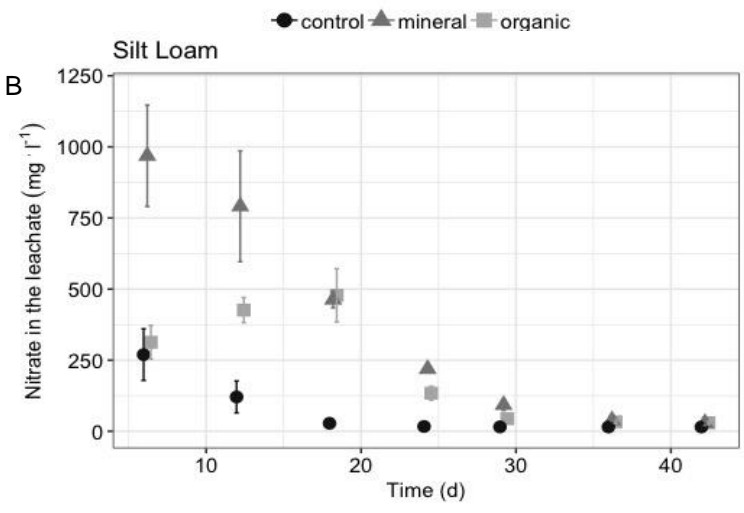

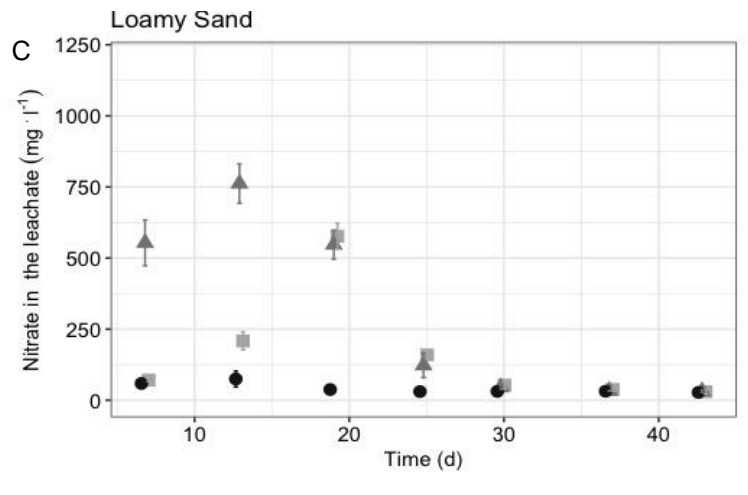

**Figure 2: Mean (± SD) nitrate (NO$_3^-$) of the soil leachate: A: Mean (± SD) totally NO$_3^-$ amount: a summation of all time series within each treatment (left: Loamy Sand; right: Silt Loam); statistical significance of difference between treatments are indicated with letters (Tukey; p < 0.05; Silt loam: n=21, Loamy Sand: n=28). B+C: temporal response curve of the mean (± SD) NO$_3^-$ concentration in the soil leachate percolating through Silt Loam (B, n= 3) and Loamy Sand (C, n=4). Shapes indicating different fertilizer application: control (circle); mineral (triangle) and organic (square).**



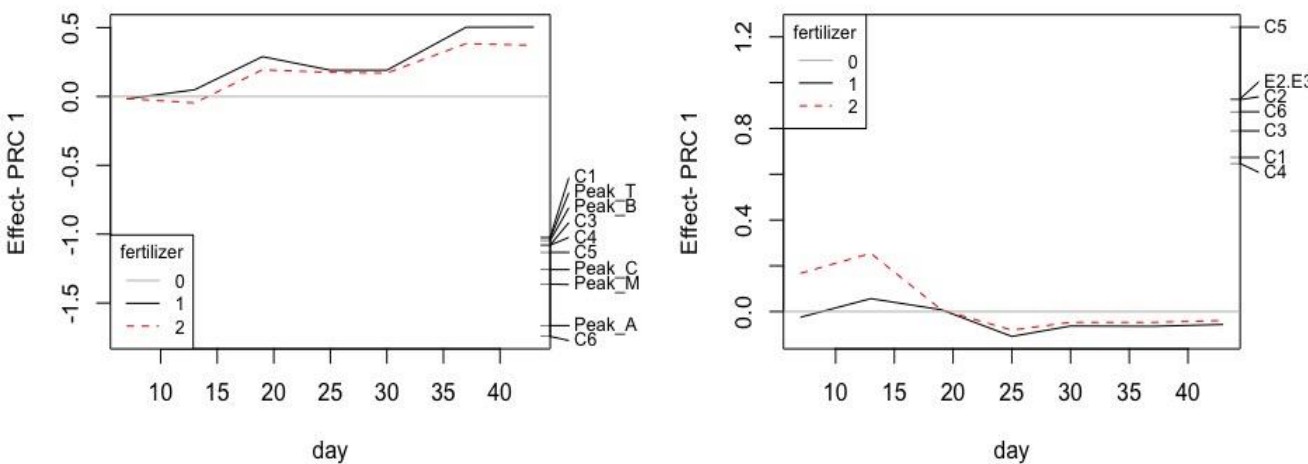

**Figure 3: left: The first component of the PRC of the Silt Loam DOM data set, using the control soil core as an internal reference (indicated by 0). 40% of the total variation of indices describing the DOM could be attributed to time (day) variation, 19% residual variance and the other 41% to between- day variation; 93% of the latter is displayed in the figure left. right: The first component of the PRC of the Loamy Sand DOM data set, using the control soil core as an internal reference (0). 86% of the total variation of indices describing the DOM could be attributed to time (day) variation, 4% as residual variance and the other 10% to between- day variation; 76% of the latter is displayed in the figure A. The species weights shown in the right part of each PCR-Figure represent the affinity of each indices with the response shown in each figure. For the sake of clarity, only species with a species weight larger than 0.5 or smaller than -0.5 are shown. The effect of DOM to fertilization is indicated by the resulting response curves (1= organic; 2= mineral).**




**Figure 4: Effects of fertilizer type on dissolved organic matter (DOM) composition percolating through Silt Loam (left) and loamy Sand (right). Mean (±SD) values of indices describing the size of the molecule (E2:E3; first line), humic-like component (C1; second line), protein- like component (C5, third line) and the degree of microbial activity (BIX; last line) over time are displayed (silty loam: n=3; loamy sand: n=4).**



725

Table 1: Basic soil properties of the soil cores for the lysimeter set-up (before refers to values prior to experiment, after refers to values after experiment; control = without fertilization, mineral = mineral fertilizer, organic = organic fertilizer). Mean (± SD), (n=3).

| Parameter | Unit | Silt Loam | | Loamy Sand | |
|---|---|---|---|---|---|
| | | 0-8 cm | 8-16 cm | 0-8 cm | 8-16 cm |
| Water content | % Mass | 18.9 ± 1.2 | 28.3 ± 1.7 | 14.3 ± 0.1 | 12.0 ±0.6 |
| pH (CaCl$_2$) | - | 6.6 ± 0.0 | 6.6 ± 0.0 | 7.6 ± 0.0 | 7.6 ±0.0 |
| TOC[before] | % C | 1.4 ± 0.0 | 1.5 ± 0.0 | 0.8 ± 0.1 | 0.6 ±0.2 |
| TOC control[after] | M% | 1.4 ± 0.0 | 1.4 ± 0.1 | 0.5 ± 0.1 | 0.5 ± 0.1 |
| TOC mineral[after] | M% | 1.4 ± 0.1 | 1.4 ± 0.1 | 0.5 ± 0.1 | 0.4 ± 0.2 |
| TOC organic[after] | M% | 1.4 ± 0.1 | 1.3 ± 0.0 | 0.5 ± 0.1 | 0.4 ± 0.1 |
| CaCO$_3$[befor] | M % CaCO$_3$ | <0.9 | <0.9 | 20.5 ± 0.1 | 21.8 ± 0.9 |
| CaCO$_3$ control[after] | M% | <0.9 | <0.9 | 21.9 ± 0.9 | 21.9 ± 1.2 |
| CaCO$_3$ mineral[2] | M% | <0.9 | <0.9 | 21.5 ± 0.3 | 22.4 ± 1.7 |
| CaCO$_3$ organic[2] | M% | <0.9 | <0.9 | 21.8 ± 0.4 | 22.1 ± 0.8 |
| Bulk density | g · cm$^{-3}$ | 1.28 ± 0.05 | 1.40 ± 0.02 | 1.39 ± 0.01 | 1.44 ± 0.00 |
| Coarse material | % | - | - | 0.1 ± 0.0 | 0.1 ± 0.0 |
| Sand | % | 6.6 ± 0.1 | 6.3 ± 0.5 | 54.8 ± 0.7 | 57.7 ± 2.8 |
| Silt | % | 65.2 ± 0.4 | 65.2 ± 0.2 | 35.5 ± 0.6 | 34.3 ± 1.9 |
| clay | % | 28.2 ± 9.5 | 28.5 ± 9.5 | 9.6 ± 0.2 | 7.9 ± 0.9 |





730

**Table 2: Statistical results testing the effects of soil texture (soil), fertilizer type and time (days) on the concentration of leached nutrients (TOC, DOC, NO$_3^-$) as well as on the optical properties of the leached organic matter (DOM composition) by using repeated measures MANOVA and permutational MANOVA (PERMANOVA) are given. (Significant effects in bold).**

| | Repeated measures MANOVA | | | | | | PERMANOVA | |
|---|---|---|---|---|---|---|---|---|
| | **TOC** | **DOC** | **NO$_3^-$** | **Cl$^-$** | **EC** | **pH** | **DOM composition** | |
| | p- value | p- value | p- value | p-value | p-value | p- value | p- value | R$^2$ |
| Soil texture | **<0.001** | **0.009** | **0.001** | **<0.001** | **<0.001** | **<0.001** | **<0.001** | 0.132 |
| Fertilizer type | **<0.001** | **<0.001** | **<0.001** | **<0.001** | **<0.001** | 0.093 | **<0.001** | 0.085 |
| days | **<0.001** | **<0.001** | **<0.001** | 0.673 | **<0.001** | **<0.001** | **<0.001** | 0.213 |
| Soil * fertilizer type | 0.011 | **<0.001** | 0.636 | 0.673 | 0.457 | 0.912 | **<0.001** | 0.067 |
| Soil * days | **<0.001** | **<0.001** | **<0.001** | **<0.001** | **<0.001** | **<0.001** | **<0.001** | 0.275 |
| Fertilizer * days | **<0.001** | **<0.001** | **<0.001** | **<0.001** | **<0.001** | 0.256 | **<0.001** | 0.066 |
| Soil * Fertilizer type * Days | **<0.001** | **<0.001** | 0.028 | 0.232 | 0.146 | 0.777 | **<0.001** | 0.042 |
| | | | | | | | Residuals | 0.119 |

TOC: Total organic carbon; DOC: dissolved organic carbon; NO$_3^-$: nitrate; Cl$^-$: Chloride; EC: electrical conductivity

735



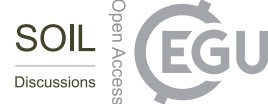

**Table 3: Six components (C1- 6) are identified by PARAFAC modeling. Range and maximum of emission and excitation wavelengths (nm) are given; secondary peaks are displayed in brackets; including previous identification by literature**

| Component | Emission (nm) | Excitation (nm) | Beforehand identified as: |
|---|---|---|---|
| 1 | 430- 500 460 | 240- 380 245 (360) | Humic- like (Coble, 1996; Graeber et al., 2012; Williams et al., 2010); oxidized quinone- like (Chai et al., 2019); Terrestrial or ubiquitous (Coble, 1996; Williams et al., 2010) derived from OM from plant and soil (Ziegelgruber et al., 2013) HMC |
| 2 | 375- 545 425 | 240- 320 <240 | Humic- like (Graeber et al., 2012; Williams et al., 2010) terrestrial OM products, mainly forest and wetlands (Stedmon and Markager, 2005) oxidized quinones- like (Cory and McKnight, 2005) HMC |
| 3 | 350- 450 390 | 240- 360 <240 (320) | Humic – like (Graeber et al., 2012; Romero et al., 2017; Williams et al., 2010); terrestrial (Stedmon and Markager, 2005) LMW fulvic/humic (Olefeldt et al., 2013) microbially altered DOM (Tfaily et al., 2015) |
| 4 | 450- 530 510 | 250- 430 260 (410) | Close to humic- like (Cory and McKnight, 2005; Stedmon and Markager, 2005), soil fulvic acid- like (Chai et al., 2019; Stedmon et al., 2003), ubiquitous (Stedmon and Markager, 2005), reduced terrestrial semi- quinone- like (Cory and McKnight, 2005), positively related to agriculture and bacterial production (Graeber et al., 2012) |
| 5 | 310- 400 320 | 240- 330 280 (240) | Tryptophan- like (Gao et al., 2016; Graeber et al., 2012), microbial processing; Protein- tannin- like (Romero et al., 2017), amino acids, free or bound in proteins; indicate intact proteins or less degraded peptide material (Fellman et al., 2010) |
| 6 | 280- 520 510 | 260- 280 260 | Soil fulvic acid- like; microbial transformed (Williams et al., 2010) |