# Peer review of "Short-term effects of fertilization on dissolved organic matter (DOM) in soil leachate"

_SOIL, 2019_

## Referee Comment (RC1) · Anonymous Referee #1 · 29 Jan 2020

This manuscript reports a study on the effects of calcium ammonium nitrate and pig slurry on the composition and concentration of dissolved organic matter in two soils. In general, the manuscript is well structured and written. However, there are many studies in the literature covering this topic and using similar approaches (e.g., Tye and Lapworth, 2016; Li et al., 2019; Seifert et al., 2016), some cited by the authors. Other major concern is that only two soils were used. This hardly represents a wide variety of soil textures and makes the results hardly generalizable.

Technical and typographical comments and suggestions

Title. Remove "(DOM)." No need to define or use abbreviations in the title.

L. 27. "highly dependent on"

[Figure]

L. 28. This last sentence is not clear. Please clarify.

L. 38. "generously dimensioned fertilizer application rates" Please reword.

L. 46. Please clarify what you mean by "light availability."

L. 58-59. Why are these results contradictory?

L. 73. "... water flow." Provide a reference.

L. 87-89. If the novelty of the study relies on the use on undisturbed soil cores, the results should have been compared to those obtained using disturbed cores.

L. 90. The reasons for these hypotheses need to be better explained (e.g., why do you expect higher DOC concentrations in coarser soils?).

L. 114. Clarify how the different sampling times could have affected the results.

L. 133. Why this specific temperature and this specific humidity?

L. 133-134. The authors convey in the introduction that they intend to make the study as realistic as possible (L. 87). So why keeping the soils in the dark?

L. 156. Soil organic C analysis needs to be conducted on finely ground samples (not on 2-mm-sieved soils).

L. 182-195. Were inner filter effects considered for calculations based on fluorescence?

L. 200. Check the R version used (the version reported seems too old for 2019).

L. 225 I suggest reporting the exact p-values.

L. 231. Report in mg per g of soil, or similar.

L. 317. This is surprising indeed. The generalizability of these findings and their implications needs to be better discussed.

L. 373. "higher microbial activity." This is not composition as reveled by fluorescence.

Please reword.

L. 377-378. I don't follow this explanation. Microbial decomposition should degrade fist the most labile compounds, thus leaving behind highly recalcitrant organic matter, with high molecular weight and aromaticity.

L. 392. What do you mean by "harmonizing DOM composition"?

L. 398. Remove "even."

L. 415. But your results suggest just the opposite, that agriculturally derived DOM will decrease in aquatic ecosystems.

L. 424-425. Again your results suggest the opposite.

It would be beneficial to show some representative fluorescence spectra.

---

## Referee Comment (RC2) · Anonymous Referee #2 · 6 Mar 2020

In this manuscript, the results of a leaching experiment using undisturbed soil columns were presented. The authors aimed to reveal the effects of mineral and organic fertilization on amounts and composition of DOM in dependence on soil texture. There are many published results available in the literature showing quite contrasting results related to this topic. I do not see that the results of the present study increase our knowledge about effects of fertilization or soil texture on DOM leaching. The use of only two different soils does not allow general conclusions.

In addition to the limited contribution of the study to increase our knowledge about dynamics of DOM in agricultural soils and their controls, the experimental approach is not convincing. First, the authors compared effects of fertilization in dependence on soil texture using two soils sampled at different seasons. One soil was samples in

early spring whereas the other soil was sampled in fall. Taking into account the strong seasonality of DOM dynamics in soils (amounts and composition) it is very difficult to compare the results of the two soils. Second, the authors did not give any information about the used pig slurry, e.g. DOC concentration, DOM composition. Therefore, it is impossible to assess effects of organic fertilization if amounts and composition of added organic matter is not known. In addition, the studied two soils differ in pH (1 unit). A higher pH should be related to a higher solubility and could superimpose assumed effects of fertilization and soil texture.

I have some furhter comments related to the methodological approach and the text of the manuscript: Line 25: The authors did not determine molecular size – the methods they used might indicate some changes occurring during the experiment. Line 64: The term stabilization is used in two different aspects; (i) solubility / desorption and (ii) decreasing microbial decay Line 93: Justification of the hypothesis is needed. Line 109: The combination of a loamy sand, 0.8% of organic C and a Haplic Chernozem is not a very common one. Line 185: The authors did not describe any correction of the fluorescence spectra, e.g. for inner filter effects. In their figures, the authors mixed up soil type and soil texture.

Although the topic of this study might be of high interest for the readership of SOIL, I cannot recommend publishing because of the indicated weaknesses and the limited new knowledge they provided. The authors might think about the main story they want to tell. Then they might consider re-writing the manuscript as a short communication taking the limited character of their study into account.

---

## Author Comment (AC1) · 7 Apr 2020

We would like to thank the reviewer for her/his critical, concise and conscientious comments, which helped to improve the quality of this manuscript. Our response follows the outline of the comments given by the reviewer.

General comments: Referee:This manuscript reports a study on the effects of calcium ammonium nitrate and pig slurry on the composition and concentration of dissolved organic matter in two soils. In general, the manuscript is well structured and written. However, there are many studies in the literature covering this topic and using similar approaches (e.g., Tye and Lapworth, 2016; Li et al., 2019; Seifert et al., 2016), some cited by the authors. Other major concern is that only two soils were used. This hardly

represents a wide variety of soil textures and makes the results hardly generalizable.

Answer: Most studies evaluating the effects of fertilization on dissolved organic carbon (DOC) concentrations and dissolved organic matter (DOM) composition have been based on the analysis of water extractable soil organic matter (WEOM) in disturbed soil samples (De Troyer et al., 2011; Gao et al., 2016; Gonet and Debska, 2011; Lei et al., 2017; Seifert et al., 2016; Tye and Lapworth, 2016; Wang et al., 2016; Xu et al., 2018; Zhang et al., 2019). However, due to the destructive sampling, the WEOM method may overestimate the immobile fraction of organic matter, which is permanently or temporarily sorbed onto soil particles (Guigue et al., 2015). Deviations between the WEOM content and the actually leached DOC concentrations have been already demonstrated with zero tension lysimeters by Lu et al. (2013). In addition, the lack of a standardised method to acquire WEOM is challenging. "This is crucial since WEOM is affected by several factors, such as the composition of the extractant (deionised water or diluted salt solution), soil pre-treatment (fresh or air-dried), soil solution ratios, and the contact time (De Feudis et al., 2017)- direct quote L356- 358". We have addressed this issue in L 83- 88 and L355- 366 of our manuscript, respectively.

To overcome these problems, we decided to use undisturbed soil samples in a controlled scientific laboratory experiment with microlysimeters. The complexity of the experimental setup limited the number of possible soils/replicates. However, we are convinced that using undisturbed soil samples in a laboratory experiment provided a more realistic picture of the soil leaching processes, while the controlled conditions in the laboratory facilitated the understanding of the underlying mechanisms. Because of the limited number of soils that could be tested, we decided to focus on two soils differing in soil texture, as literature has shown that soil texture is a driving factor for leaching (Autio et al., 2016). We selected silt loam , because it is one of the most dominating textures in European agriculture (Ballabio et al., 2016), and compared this with a loamy sand because of its different soil physical properties.

A number of studies assessed the effects of fertilizer on organic matter cycling and

organic carbon leaching either with mineral (Fröberg et al., 2013; Liang et al., 2016; Lu et al., 2013; McDowell et al., 2004; Sjöberg et al., 2003; Tian et al., 2017) or organic fertilizers (Adams et al., 2005; Lloyd et al., 2012; Long et al., 2015). However, only few studies compared the effects of both mineral and organic fertilizers on organic carbon leaching directly in the same experiment (Gonet and Debska, 2011; Manna et al., 2005; Xu et al., 2018). Our study demonstrates that organic and inorganic fertilizer show partly similar and partly deviating effects on the leaching due to the provision of either nutrients alone or nutrients plus organic matter. We consider this a crucial information for further soil leaching studies. Thus, we think that our study will contribute significantly to the mechanistic understanding of DOC leaching from undisturbed soils due to the four-factorial design of different soil textures and different fertilizer types, even though the number of tested soils is restricted. However, the comments of both reviewers had shown us that we need to stress the significance of our study and the knowledge gain for the scientific community more clearly in a revised version.

Detailed comments on the manuscript: Referee: Remove "(DOM)." No need to define or use abbreviations in the title. Answer: We will remove the abbreviation "(DOM)" in the title as suggested.

Referee: L. 27. "highly dependent on" Answer: Original: Furthermore, the magnitude of fertilization on DOC concentrations and DOM composition was highly depending on the soil texture they originate from. Proposed modification: Furthermore, the magnitude of fertilization on DOC concentrations and DOM composition was interrelated with soil texture.

Referee: L. 28. This last sentence is not clear. Please clarify. Answer: Original: At ecosystem level, a sufficiently long soil passage might mitigate the impact of fertilization on soil DOM. Proposed modification: Due to sorption processes and microbial decomposition throughout the soil profile, soil passage may overlay the fertilizer-induced changes of DOM composition.

Referee: L. 38. "generously dimensioned fertilizer application rates" Please reword. Answer: Original: Decades of generously dimensioned fertilizer application rates have increased the overall nitrogen content in soils, thus... Proposed modification: Decades of high fertilizer application rates have increased the overall nitrogen content in soils, thus..

Referee: L. 46. Please clarify what you mean by "light availability." Answer: Original: "..and changes of in-stream microbial processes resulting from increased nutrient and light availability" Proposed modification: In order to clarify the above-mentioned statement, we propose to add the following sentence: "Deforestation of riparian forests by farmers increases the light intensity in fluvial systems and, thus, stimulates algal growth, which in turn alters the DOM composition."

Referee: L. 58-59. Why are these results contradictory? Answer: Original: "Intensification of agriculture tends to increase protein-like, labile, and microbially-transformed organic matter with low redox states and low molecular weight and aromaticity (Wu et al., 2019b; Hosen et al., 2014; Wilson and Xenopoulos, 2009). However, in some studies, organic matter in agricultural streams exhibited highly complex structure with a high level of humified and microbial-derived DOM (Graeber et al., 2012; Heinz et al., 2015). These partly contradicting results arise...."

Explanation: Graeber et al., 2012 and Heinz et al., 2015 found a higher degree of humified DOM in agricultural streams (humic-like DOM), while the others discovered increased proportions of fresh, microbial-derived DOM with a low degree of humification in agricultural influenced streams (Wu et al., 2019b; Hosen et al., 2014; Wilson and Xenopoulos, 2009). The reasons for these different results lie in the high number of different influencing factors and the actual situation of the studied streams (e.g. land use, agricultural practices, soil properties, climate, etc). We have already mentioned this in the original manuscript (L59-61), but will address it more clearly in a revised version (see proposed modifications below).

[Figure]

Proposed modification: "Intensification of agriculture tends to increase protein-like, labile, and microbially-transformed organic matter with low redox states and low molecular weight and aromaticity (Wu et al., 2019b; Hosen et al., 2014; Wilson and Xenopoulos, 2009). Authors have explained this by the enhanced nutrient availability stimulating autochthonous DOM production and decreased water retention times in the soils due to tillage, amongst others. However, in some studies, organic matter in agricultural streams exhibited a high level of humification and a low redox state (Graeber et al., 2012; Heinz et al., 2015). These findings indicate that aquatic DOM is shaped by the terrestrial DOM source, whereby its composition depends on the specific land use, soil properties, agricultural management practices, the flow paths, by which the DOM is transported to the stream (e.g., overland flow, groundwater, drainage water), and the various in-stream microbial transformation processes (Stanley et al., 2012)."

Referee: L. 73. "... water flow." Provide a reference. Answer: Proposed modification: Furthermore, leaching of nutrients and organic carbon is directly proportional to water flow. (Fei et al., 2019).

Referee: L. 87-89. If the novelty of the study relies on the use on undisturbed soil cores, the results should have been compared to those obtained using disturbed cores. Answer: The main aim of our study was to investigate the effects of different fertilization on DOM leaching in undisturbed soils and not to investigate the effects of disturbance (disturbed vs. undisturbed soil cores). We used undisturbed soil cores to avoid effects that may have been caused by disturbance as suggested by available literature.

Referee: L. 90. The reasons for these hypotheses need to be better explained (e.g., why do you expect higher DOC concentrations in coarser soils?). Answer: Original: We hypothesized that: (1) Both, mineral and organic fertilization will increase the concentrations of leached dissolved organic carbon (DOC) in the soil pore water and will shift DOM composition towards more labile compounds. (2) Pore water of coarser textured loamy sand will exhibit higher DOC concentrations than finer textured silt loam soil.

Proposed modification: We hypothesized that: 1) Fertilization fuels mineralization in the soil, which will in turn increase the DOC concentration in the soil pore water percolating through the soil. Additionally, fertilization will induce a shift in the composition of DOM towards more labile compounds with lower molecular weight and less humicity. These effects will be more pronounced in the organic fertilizer application than in the mineral fertilizer, because the manure provides additional organic carbon to the soil. 2) Since soil texture is a strong predictor of DOC leaching (Autio et al., 2016), we assume that the coarser textured loamy sand soil is more prone to leaching than the finer textured silty loam soil, resulting in a rapid increase of DOC in the leachate. Due to shorter retention times and, thus, probably lower microbial decomposition, we expect that the leached DOM of the loamy sand soil will consist of a higher share of humic compounds. In a revised version, we will reorganize the introduction of this manuscript to specify the hypothesis.

Referee: L. 114. Clarify how the different sampling times could have affected the results. Answer: Due to the comprehensive experimental set-up (and sampling schedule), including two fertilizer types and a control with four replicates each, we had to perform the experiments with the two soils separately in consecutive order. This has unavoidably led to some seasonal differences between the samplings, e.g. with regard to the microbial activities. In order to minimize eventual seasonal effects, the soil cores were acclimated under laboratory conditions 7 days before the start of the experiment. Besides, the treatments were compared to a control to account for differences in seasons or soil properties. In addition, we tried to minimize the effects of season by facilitating similar (or maximum comparable) management preconditions of the soils sampled. Both soils had been last treated with mineral fertilizer (NAC 27- calcium ammonium nitrate) 5 months before the sampling. In both cases, crops had been removed between 1 to 3 months prior to the experiment, the soils had been chiselled, and the seedbeds had been prepared exactly 3 days before the sampling. We will add this information in a revised version of the manuscript.

Referee: L. 133. Why this specific temperature and this specific humidity? Answer: Original: The lysimeter experiments were conducted under controlled environmental conditions of 17°C air temperature and 78 % relative humidity. Explanation: The temperature and humidity of the laboratory environment was determined by the process constraints of the microlysimeter set-up. However, our target air temperature of > 15.6° was chosen to reflect the mean air temperature for the growing season from April till October in Lower Austria (zamg.ac.at). We will add this information in a revised version of the manuscript.

Proposed modification: The lysimeter experiments were conducted under controlled environmental conditions of 78 % relative humidity and 17°C air temperature. The latter was chosen to reflect the mean air temperature for the growing season from April till October in Lower Austria (zamg.ac.at).

Referee: L. 133-134. The authors convey in the introduction that they intend to make the study as realistic as possible Answer: Original: "The lysimeter experiments were conducted under controlled environmental conditions of 17°C air temperature and 78 % relative humidity." Explanation: In our experimental set-up, the term "as realistic as possible" refers mainly to the soil structure and pore space of an undisturbed soil and only partly to the environmental conditions (e.g. dark incubations, high humidity, and moderate air temperatures; see L86-88). We will explain the term " as realistic as possible" more detailed in a revised version of the manuscript with specific reference to soil structure to avoid misunderstandings.

Referee: L. 87. So why keeping the soils in the dark? Answer: Darkness was chosen to avoid photodegradation of DOM and algal growth in the soil leachate sample.

Referee: L. 156. Soil organic C analysis needs to be conducted on finely ground samples (not on 2-mm-sieved soils). Answer: Soil organic carbon was determined by dry combustion, following the OENORM L 1080 (the Austrian normative on chemical analysis of soil organic carbon TOC) and EN ISO 10 694, respectively. Therefore, soil

samples were air- dried and sieved to 2 mm.

Referee: L.182-195. where inner filter effects considered for calculations based on fluorescence? Answer: Yes, the pre-processing of the dataset was done with the "staRdom" package built for R (Pucher et al., 2019). Pre-processing of the dataset included: blank subtraction, inner-filter effect correction and the removal and interpolation of Rayleigh and Raman scattering of first and second order. In addition, the outcomings of the PARAFAC- analysis were evaluated with a split-half analysis and the obtained model was evaluated using correlations between the component (Murphy et al., 2013). We will include this information in a revised version.

Referee: L. 200. Check the R version used (the version reported seems too old for 2019). Answer: The whole statistical analysis was done with R (3.5.0). This version was chosen to run the package "staRdom".

Referee: L. 225 I suggest reporting the exact p-values. Answer: Proposed modification: We will include exact p-values in a revised version

Referee: L. 231. Report in mg per g of soil, or similar. Answer: Original: "Average amounts of leached DOC ... ranged between 20.4 mg (silt loam, mineral fertilizer) and 34.4 mg (....)..." These results reported the totally leached DOC throughout the experiment for the soil core. Proposed modification: We have added the term "amount for 3 times exchanged pore volume" to text.

Referee: L. 317. This is surprising indeed. The generalizability of these findings and their implications needs to be better discussed. Answer: Original: "In contrast to our original expectations of enhanced DOC leaching from fertilized soils, fertilization reduced DOC concentrations of the leachate during the first 2-3 weeks. The only exception was organic fertilization of a loamy sand, where similar DOC concentrations as in the control were observed throughout the whole experiment."

Explanation: We totally agree with the reviewer that these results are not only surprising, but that they may provide new insights into the underlying mechanisms of the effects of fertilization on the DOC leaching from undisturbed soils, highlighting the significance of our study for future research. We have tried to explain the mechanisms leading to these results in the discussion following this line, but will improve this section in a revised manuscript by disentangling the different, interacting mechanisms more clearly. Basically, we have identified three different mechanisms behind the effects of fertilization on the DOC leaching in undisturbed soils: 1. Nutrients (nitrate) stimulate the mineralization of the soil organic matter and transformation of OC to CO2, resulting in a decreased leaching of DOC. Evidence: the reduction of DOC leaching disappears with the decrease in the nitrate leaching (this study) and other studies about inorganic fertilizers in undisturbed soils show similar results (e.g. Lu et al. 2013; Sjöberg et al. 2003). Currently referred to in the manuscript in L325-332. 2. Organic fertilizer adds organic carbon to the soil, thus potentially reducing or even compensating the above mentioned effects of enhanced mineralization by nutrients. Evidence: the reduction of DOC leaching is smaller in the organic fertilizer treatment than in the inorganic fertilizer treatment (this study) and other studies about organic fertilizer in undisturbed soils show similar results (no effects or increased DOC leaching; see e.g. Long et al 2015, Lloyd et al 2012). Currently referred to in the manuscript in L332-L337. 3. The water retention time, determined by the soil texture, influences the amount of soil processing (mineralisation) versus transportation; thus, in the loamy sand with organic fertilizer, DOC leaching prevails; Currently discussed in the manuscript in L340-351.

Referee: L. 373. "higher microbial activity." This is not composition as revealed by fluorescence. Please reword. Answer: Original: ". . . revealed a high aromaticity (Fi), a higher microbial activity (BIX) and a smaller molecular size (E2:E3) . . .."

Proposed modification: Initial DOM originating from fertilized silt loam soil revealed a higher proportion of microbial derived DOM (Fi), a high proportion of freshly produced DOM (BIX) and a smaller molecular size (E2:E3)..

Referee: L. 377-378. I don't follow this explanation. Microbial decomposition should

degrade fist the most labile compounds, thus leaving behind highly recalcitrant organic matter, with high molecular weight and aromaticity. Answer: Original: We assume that fertilization fueled the microbial decomposition of the soil organic matter, thus shifting DOM towards more labile compounds with low molecular weight and lower aromaticity.

Explanation: Generally, two contrasting processes occur during microbial decomposition of soil organic matter: Soil microbes decompose labile organic matter, this leads to an accumulation of recalcitrant organic matter. In opposition to this, nutrients applied through fertilization may promote the decomposition of complex (recalcitrant) soil organic matter pools, thus stimulating both, the production and leaching of fresh organic matter with a high biodegradability. The latter could be proven in our experiments, when soil was fertilized with the mineral fertilizer. Laboratory incubations and field mesocosms studies have shown that bacterial mineralization can be both a source or a sink for labile substances, such as e.g. amino acids (Coble et al., 2014; Moran et al., 2000; Stedmon and Markager, 2005; Yamashita and Tanoue, 2003).

Referee: L. 392. What do you mean by "harmonizing DOM composition"? Answer: Original: However, other studies showed that long-term fertilization can modify soil microbial decomposer communities, thus overlaying the effects of soil distinct properties and harmonizing DOM composition of soil humic substances (Li et al., 2019; Wu M. et al., 2019a). Proposed modification: However, other studies showed that long-term fertilization can lead to shifts in microbial decomposer communities, thus resulting in similar DOM compositions across soils and overlaying distinct soil properties (Li et al., 2019; Wu M. et al., 2019a).

Referee: L. 398. Remove "even." Answer: Proposed modification: We will remove the word 'even' in the text.

Referee: L. 415. But your results suggest just the opposite, that agriculturally derived DOM will decrease in aquatic ecosystems. Answer: Original: Due to the spatial extent

of agricultural land use and the yearly growing application of nitrogen-based fertilizers, it is reasonable to assume that the share of agricultural derived DOM will increase in terrestrial and aquatic ecosystems.

Proposed modification: Due to the predicted intensification of agricultural land use, it is reasonable to assume that an increasing proportion of terrestrial DOM inputs to aquatic systems will originate from agricultural areas.

Explanation: This statement referred to the general development of agriculture and the proportion of agricultural derived DOM compared to the total terrestrial DOM imported into aquatic systems. We have slightly rephrased it to avoid misunderstandings. However, our study does definitely not show that agriculturally derived DOM will decrease in aquatic ecosystems. It just emphasizes that soils have a huge potential to buffer fertilization effects (also from organic fertilizer, which is per se an additional OM source), if there is sufficient time for DOM and nutrient uptake during the soil passage. This highlights the importance of long residence times in terrestrial systems and the need for management measures, which prevent or reduce fast flow paths leading soil water directly into aquatic systems, such as surface flow, fast subsurface flow, or drainage water. Again, this is a conclusion, which is only possible due to our approach of using undisturbed soil columns.

Referee: L. 424-425. Again your results suggest the opposite. It would be beneficial to show some representative fluorescence spectra. Answer: Original: On a global perspective, intensification of agricultural management may favour the leaching of biogeochemically reactive DOM to aquatic ecosystems. Explanation: In order to avoid misunderstandings, we have rephrased the conclusion:

Proposed modification: Conclusion Due to the spatial extent of agricultural land use and the yearly growing application of nitrogen-based fertilizers, it is reasonable to assume that an increasing proportion of terrestrial DOM inputs to aquatic systems will originate from agricultural areas. Thus, understanding the diverse effects of agricultural practices, such as, e.g., fertilization, tillage or harvesting, on the export of DOC from different types of soils under different climatic conditions is crucial to protect and sustainably manage freshwater systems in agricultural landscapes. Our experiments with undisturbed soils demonstrate that the effects of fertilization on soil DOC leaching depend on the fertilizer type and are strongly interrelated with soil texture. This has implications on both the amount and the quality of the leached DOC. In our study, fertilization tended to increase the proportion of fresh, microbially transformed DOM, while it generally reduced the amount of leached DOC compared to untreated soils probably due to the stimulation of microbial mineralisation of soil organic matter through the enhanced nutrient supply. This effect was more pronounced in the silt loam soil than in sandy loam soil. Consequently, it is possible to deduce that a longer residence time of leachate in soils can reduce the DOC exports from soils into streams. This implies that increased DOC concentrations in agriculturally influenced streams do probably not originate from soil pore water, but from faster flow paths, such as surface runoff (overland flow), subsurface runoff or drainage water. Further studies are needed to identify the main sources and pathways of terrestrial DOC inputs to stream systems and analyse the underlying mechanisms controlling the relation between DOC transport and processing in the soil. In future, many regions will become dry during summer with more precipitation per rainfall event due to global warming. High precipitation rates on dry soils usually generate a rapid surface runoff with a high sediment load. Therefore, we assume that more agriculturally derived DOM will be entering aquatic systems without a sufficient soil passage in the future. Overall, our results have further implications for organic carbon and organic matter management. Due to the complexity of agroecosystems and the resulting myriad effects on organic carbon processing, this will require further investigations.

References: Adams, A.B., et al (2005). Forest Ecology and Management 220, 313–325. Autio, I., et al (2016). Ambio 45, 331–349. Ballabio, C., et al (2016). Geoderma 261, 110–123. Coble, P., et al (2014). Cambridge University Press, Cambridge. De Feudis, M., et al (2017). Geoderma 302, 6–13. De Troyer, I., et al (2011). Soil Biology

and Biochemistry 43, 513–519. Fröberg, M., et al (2013). Geoderma 200–201, 172–179. Gao, S.-J., et al (2016). Journal of Analytical Methods in Chemistry 2016, 1–10. Gonet, S.S., et al (2011). Plant, Soil and Environment 52, 55–63. Guigue, J., et al (2015). Soil Biology and Biochemistry 84, 158–167. Lei, Z., et al (2017). Forests 8, 452. Li, X.-M., et al (2019). Environmental Science & Technology 53, 50–59. Liang, F., et al (2016). Scientific Reports 6. Lloyd, C.E.M., et al (2012). Organic Geochemistry 43, 56–66. Long, G.-Q., et al (2015). Soil and Tillage Research 146, 270–278. Lu, X., et al (2013). Biogeosciences 10, 3931–3941. Manna, M.C., et al (2005). Field Crops Research 93, 264–280. McDowell, W.H., et al (2004). Forest Ecology and Management 196, 29–41. Moran, M.A., et al (2000). Limnology and Oceanography 45, 1254–1264. Murphy, K.R., et al (2013). Analytical Methods 5, 6557. Pucher, M., et al (2019). Water 11, 2366. Seifert, A.-G., et al (2016). Science of The Total Environment 571, 142–152. Sjöberg, G., et al (2003). Soil Biology and Biochemistry 35, 1305–1315. Stedmon, C.A., et al (2005). Limnology and Oceanography 50, 686–697. Tian, D., et al (2017). Science of The Total Environment 607–608, 1367–1375. Tye, A.M., et al (2016). Agriculture, Ecosystems & Environment 221, 245–257. Wang, S., et al (2016). Scientific Reports 6. Wu, M., et al (2019). Soil and Tillage Research 186, 105–111. Xu, P., et al (2018). Journal of Soils and Sediments 18, 1865–1872. Yamashita, Y., et al (2003). Marine Chemistry 82, 255–271. Zhang, Y., et al (2019). Environmental Science and Pollution Research.

---

## Author Comment (AC2) · 7 Apr 2020

We would like to thank the anonymous reviewer for her/his critical comments, which will help to improve the quality of this manuscript. Our response follows the outline of the comments given by the reviewer.

General comments: Referee: In this manuscript, the results of a leaching experiment using undisturbed soil columns were presented. The authors aimed to reveal the effects of mineral and organic fertilization on amounts and composition of DOM in dependence on soil texture. There are many published results available in the literature showing quite contrasting results related to this topic. I do not see that the results of the present study increase our knowledge about effects of fertilization or soil texture on

[Figure]

DOM leaching. The use of only two different soils does not allow general conclusions.

Answer: Most studies evaluating the effects of fertilization on dissolved organic carbon (DOC) concentrations and dissolved organic matter (DOM) composition have been based on the analysis of water extractable soil organic matter (WEOM) in disturbed soil samples (De Troyer et al., 2011; Gao et al., 2016; Gonet and Debska, 2011; Lei et al., 2017; Seifert et al., 2016; Tye and Lapworth, 2016; Wang et al., 2016; Xu et al., 2018; Zhang et al., 2019). However, due to the destructive sampling, the WEOM method may overestimate the immobile fraction of organic matter, which is permanently or temporarily sorbed onto soil particles (Guigue et al., 2015). Deviations between the WEOM content and the actually leached DOC concentrations have been already demonstrated with zero tension lysimeters by Lu et al. (2013). In addition, the lack of a standardised method to acquire WEOM is challenging. "This is crucial since WEOM is affected by several factors, such as the composition of the extractant (deionised water or diluted salt solution), soil pre-treatment (fresh or air-dried), soil solution ratios, and the contact time (De Feudis et al., 2017)- direct quote L356- 358". We have addressed this issue in L 83- 88 and L355- 366 of our manuscript, respectively.

To overcome these problems, we decided to use undisturbed soil samples in a controlled scientific laboratory experiment with microlysimeters. The complexity of the experimental setup limited the number of possible soils/replicates. However, we are convinced that using undisturbed soil samples in a laboratory experiment provided a more realistic picture of the soil leaching processes, while the controlled conditions in the laboratory facilitated the understanding of the underlying mechanisms. Because of the limited number of soils that could be tested, we decided to focus on two soils differing in soil texture, as literature has shown that soil texture is a driving factor for leaching (Autio et al., 2016). We selected silt loam, because it is one of the most dominating textures in European agriculture (Ballabio et al., 2016), and compared this with a loamy sand because of its different soil physical properties.

A number of studies assessed the effects of fertilizer on organic matter cycling and

organic carbon leaching either with mineral (Fröberg et al., 2013; Liang et al., 2016; Lu et al., 2013; McDowell et al., 2004; Sjöberg et al., 2003; Tian et al., 2017) or organic fertilizers (Adams et al., 2005; Lloyd et al., 2012; Long et al., 2015). However, only few studies compared the effects of both mineral and organic fertilizers on organic carbon leaching directly in the same experiment (Gonet and Debska, 2011; Manna et al., 2005; Xu et al., 2018). Our study demonstrates that organic and inorganic fertilizer show partly similar and partly deviating effects on the leaching due to the provision of either nutrients alone or nutrients plus organic matter. We consider this a crucial information for further soil leaching studies. Thus, we think that our study will contribute significantly to the mechanistic understanding of DOC leaching from undisturbed soils due to the four-factorial design of different soil textures and different fertilizer types, even though the number of tested soils is restricted. However, the comments of both reviewers have shown us that we need to stress the significance of our study and the knowledge gain for the scientific community more clearly in a revised version. The study can certainly not (and was never intended) to deliver generally (or widely) applicable results on DOC leaching rates or amounts due to the restricted number of specific tested soils, as the reviewer rightly point out. However, the 2x2 factorial design, with different soil types and different fertilizer types tested in undisturbed soils under controlled experimental lab conditions very well facilitates general conclusions about mechanisms behind the interaction of soil texture and fertilizer type, which provides valuable information for further research and is applicable to other scientific studies. Our results may encourage other colleagues to focus more on soil processes in undisturbed soils and on the interactive effects of the different drivers for soil leaching to improve our current knowledge and enable sustainable management of soils in agricultural landscapes.

Referee: In addition to the limited contribution of the study to increase our knowledge about dynamics of DOM in agricultural soils and their controls, the experimental approach is not convincing. First, the authors compared effects of fertilization in dependence on soil texture using two soils sampled at different seasons. One soil was samples in early spring whereas the other soil was sampled in fall. Taking into account

the strong seasonality of DOM dynamics in soils (amounts and composition) it is very difficult to compare the results of the two soils.

Answer: While acknowledging the concerns of the reviewer about the restricted number of soils and the different sampling times, we do not agree to the argument of "limited contribution of the study to increase our knowledge" nor can we agree that "the experimental approach is not convincing". Certainly, any scientific experiment would gain in significance if the number of treatments, replicates, or influencing factors is increased. However, as stated above, not many studies have so far looked into the interactive effects of soil texture and fertilizer type in undisturbed soils under controlled lab conditions; thus, we regard our study as a valid contribution to the general knowledge of fertilization effects on soil leaching of organic carbon quantity and quality. Regarding the comments about the seasonality, we agree that the different sampling times may have an influence on the results. Due to the comprehensive experimental set-up (and sampling schedule), we had to perform the experiments with the two soils separately in consecutive order. This has unavoidably led to some seasonal differences between the samplings, e.g. with regard to microbial activities. This is why we compared the fertilizer treatments with the untreated control of the same soil, but not primarily across soils. However, we tried to minimize eventual seasonal effects by several means: In order to reduce the effects of in-field conditions, the soil cores were acclimated under laboratory conditions 7 days before the start of the experiment. The treatments were compared with a control to account for differences in seasons or soil properties. In addition, we tried to facilitate maximum comparable management preconditions of the soils sampled, a factor which may cause a much higher variability than seasonal effects mentioned by the reviewer. Both soils had been treated with mineral fertilizer (NAC 27- calcium ammonium nitrate) latest 5 months before the sampling. In both cases, crops had been removed between 1 to 3 months prior to the experiment (one spring harvest, one autumn harvest), the soils had been chiselled, and the seedbeds had been prepared exactly 3 days before the sampling. Regarding the soil types, we selected two soils common for this region (the silt loam is also one of the most dominating textures in European agriculture (Ballabio et al., 2016) with clearly different soil physical properties.

Referee: Second, the authors did not give any information about the used pig slurry, e.g. DOC concentration, DOM composition. Therefore, it is impossible to assess effects of organic fertilization if amounts and composition of added organic matter is not known. In addition, the studied two soils differ in pH (1 unit). A higher pH should be related to a higher solubility and could superimpose assumed effects of fertilization and soil texture.

Answer: We would like to thank the reviewer for her/ his excellent idea. In a revised version of this manuscript we will include DOC concentration and composition as well as nutrient concentration of the used organic fertilizer, pig slurry. Indeed, soil pH affects the solubility and transportation of DOM through the soil as also other soil properties, such as texture or soil chemistry, do in nature. Our study design is intended to reflect the real situation in the field as best as possible (for this region), including the natural differences in the properties of the tested soils. We consider the inclusion of this variability an important factor in the design to elucidate general mechanisms as were found in our study, such as e.g. a potentially enhanced OM mineralization through nutrients or the interaction of DOC transport vs internal OC processing. However, we will explicitly address the influence of pH in the discussion in a revised version as suggested by the reviewer.

Proposed adaption: As suggested by the reviewer, we will add chemical characteristics of the used organic manure. Additionally, we will explicitly emphasize the impact of soil pH in the discussion in the revised version of the manuscript.

Detailed comments

Referee: Line 25: The authors did not determine molecular size – the methods they used might indicate some changes occurring during the experiment.

Answer: Information about molecular sizes were obtained via absorbance spectra and the most commonly used indices SUVA and E2:E3 (see L191-192). While this certainly does not yield the information gained by Fourier transform ion cyclone resonance mass spectrometry (FT−ICR MS; Li et al., 2019), it nevertheless is an established surrogate parameter for molecular size, which is widely used in the literature (e.g. (Avneri-Katz et al., 2017; Awad et al., 2015; D'Andrilli et al., 2019; Fleck et al., 2014; Frey et al., 2016; Graeber et al., 2015; Heinz et al., 2015; Inamdar et al., 2012; Li and Hur, 2017; Olshansky et al., 2018; Singh et al., 2018; Weishaar et al., 2003; Xian et al., 2018; Xu and Guo, 2017). In fact, our study actually showed a change in SUVA and E2:E3 during the experiments, indicating changes in the molecule sizes (e.g. L296-298; L309-311).

Referee: Line 64 :The term stabilization is used in two different aspects; (i) solubility / desorption and(ii) decreasing microbial decay

Answer: Original: "Stabilization of DOM against leaching depends on the chemical nature and physical properties of sites available to interact with the ions and molecules present in soil solution (Baldock and Skjemstad, 2000; Silveira, 2005)." Proposed modification We agree that this term is ambiguous, and we will change it in a revised version as follows: "The retention of DOM against leaching depends on the chemical nature and physical properties of sites available to interact with the ions and molecules present in soil solution (Baldock and Skjemstad, 2000; Silveira, 2005)."

Referee: Line 93: Justification of the hypothesis is needed.

Answer: Original: "Pore water of coarser textured loamy sand will exhibit higher DOC concentrations than finer textured silt loam soil." Proposed modification: Since soil texture is a strong predictor of DOC leaching (Autio et al., 2016), we assume that the coarser textured loamy sand soil is more prone to leaching than the finer textured silty loam soil, resulting in a rapid increase of DOC in the leachate. Due to shorter retention times and, thus, probably lower microbial decomposition, we expect that the leached DOM of the loamy sand soil will consist of a higher share of humic compounds.

Referee: Line 109: The combination of a loamy sand, 0.8% of organic C and a Haplic Chernozem is not a very common one.

Answer: We agree. However, the second soil was chosen because of its contrasting soil physical properties to the silt loam and because we had detailed information about the agricultural preconditions there, which were highly comparable to the second sampling site. The silt loam was selected, because it is one of the most dominating textures in European agriculture (Ballabio et al., 2016). In a revised version of the manuscript, we will reassess the soil type to avoid misunderstandings.

Referee: Line 185: The authors did not describe any correction of the fluorescence spectra, e.g. for inner filter effects.

Answer: We agree that this information needs to be added to a revised version, as it is crucial for assessing the quality of our data. Actually, extensive pre-processing of the DOM dataset was done with the "staRdom" package built for R (Pucher et al., 2019). Pre-processing of the dataset included: blank subtraction, inner-filter effect correction and the removal and interpolation of Rayleigh and Raman scattering of first and second order. In addition, the outcomings of the PARAFAC- analysis were evaluated with a split-half analysis and the obtained model was evaluated using correlations between the component (Murphy et al., 2013).

Referee: In their figures, the authors mixed up soil type and soil texture. Answer: Proposed Modification: We will exclude the term "soil type".

Referee: Although the topic of this study might be of high interest for the readership of SOIL, I cannot recommend publishing because of the indicated weaknesses and the limited new knowledge they provided. The authors might think about the main story they want to tell. Then they might consider re-writing the manuscript as a short communication taking the limited character of their study into account.

Answer: We respectfully disagree with the reviewer on this comment for the reasons

detailed above. To summarize: Our extensive literature research showed that fertilizer-induced changes in DOM composition and DOC leaching are still poorly explained. The data available on leaching of organic carbon in undisturbed soil columns under controlled conditions is sparse and to the best of our knowledge, there is no study investigating the immediate response of DOM leaching to organic and mineral fertilization using undisturbed soil cores. Thus, the results of this study will increase our knowledge on the behaviour and characteristics of organic carbon movement in soils.

We may add, that the experiment has been carried out to comply with scientific requirements: we performed extensive PARAFAC modelling and statistical analysis to evaluate fertilizer- induced changes in DOM composition of the soil leachate throughout the whole experiment. The precision and accuracy of the measurements were monitored continuously with four replicates per treatment. The overall, maximum allowed relative standard deviation between replicates was set to 5%. In addition, changes in DOM composition were compared with non-fertilized soil cores constantly, to avoid misinterpretation of the measurements.

Nevertheless, we are prepared to modify our manuscript in response to this comment. We will reorganize the manuscript to enhance the clarity and the focus of our work towards the effects of fertilization. We will also display half-split analysis and the outcomings of the PARAFAC in figures. We thank the reviewer for his/her comments, which will certainly help to strengthen this manuscript.

References: Adams, A.B., et al 2005. Forest Ecology and Management 220, 313–325. Autio, I., et al 2016. Ambio 45, 331–349. Avneri-Katz, S., et al 2017. Organic Geochemistry 103, 113–124. Awad, J., et al 2015. Science of The Total Environment 529, 72–81. Ballabio, C., et al 2016. Geoderma 261, 110–123. D'Andrilli, J., et al 2019. Biogeochemistry 142, 281–298. De Feudis, M., et al 2017. Geoderma 302, 6–13. De Troyer, I., et al 2011. Soil Biology and Biochemistry 43, 513–519. Fleck, J.A., et al 2014 iScience of The Total Environment 484, 263–275. https://doi.org/10.1016/j.scitotenv.2013.03.107 Frey, K.E., et al 2016. Biogeosciences

13, 2279–2290. Fröberg, M., et al 2013. Geoderma 200–201, 172–179. Gao, S.-J., et al 2016. Journal of Analytical Methods in Chemistry 2016, 1–10. Gonet, S.S., et al 2011. Plant, Soil and Environment 52, 55–63. Graeber, D., et al 2015. Hydrology and Earth System Sciences 19, 2377–2394. Guigue, J., et al 2015. Soil Biology and Biochemistry 84, 158–167. Heinz, M., et al 2015. Environmental Science & Technology 49, 2081–2090. Inamdar, S., et al 2012. Biogeochemistry 108, 55–76. Lei, Z., et al 2017. Forests 8, 452. Li, P., et al 2017. Critical Reviews in Environmental Science and Technology 47, 131–154. Li, X.-M., et al 2019. Environmental Science & Technology 53, 50–59. Liang, F., et al 2016. Scientific Reports 6. Lloyd, C.E.M., et al 2012. Organic Geochemistry 43, 56–66. Long, G.-Q., et al 2015. Soil and Tillage Research 146, 270–278. Lu, X., et al 2013. Biogeosciences 10, 3931–3941. Manna, M.C., et al 2005. Field Crops Research 93, 264–280. McDowell, et al 2004. Forest Ecology and Management 196, 29–41. Murphy, K.R., et al 2013. Analytical Methods 5, 6557. Olshansky, Y., et al 2018. Biogeosciences 15, 821–832. Pucher, M., et al 2019. Water 11, 2366. Seifert, A.-G., et al 2016. Science of The Total Environment 571, 142–152. Singh, M., et al 2018. in: Advances in Agronomy. Elsevier, pp. 33–84. Sjöberg, G., et al 2003. Soil Biology and Biochemistry 35, 1305–1315. Tian, D., et al 2017. Science of The Total Environment 607–608, 1367–1375. Tye, A.M., et al 2016. Agriculture, Ecosystems & Environment 221, 245–257. Wang, S., et al 2016. Scientific Reports 6. Weishaar, J.L., et al 2003. Environmental Science & Technology 37, 4702–4708. Xian, Q., et al 2018. Science of The Total Environment 622–623, 385–393. Xu, H., et al 2017. Water Research 117, 115–126. Xu, P., et al 2018. Journal of Soils and Sediments 18, 1865–1872. Zhang, Y., et al 2019. Environmental Science and Pollution Research.